# Female moths incorporate plant acoustic emissions into their oviposition decision-making process

Rya Seltzer[1]*[†], Guy Zer Eshel[1][†], Omer Yinon[1], Ahmed Afani[1], Ofri Eitan[1], Sabina Matveev[2], Galina Levedev[2], Michael Davidovitz[2], Tal Ben Tov[3], Gayl Sharabi[3], Yuval Shapira[3], Neta Shvil[4], Maya Harari Gibli[1], Ireen Atallah[1], Sahar Hadad[1], Dana Ment[2], Lilach Hadany[3,4][‡], Yossi Yovel[1,4]*[‡]

[1]School of Zoology, George S. Wise Faculty of Life Sciences, Tel Aviv University, Tel Aviv, Israel; [2]Plant Protection Institute, Agricultural Research Organization – Volcani Institute, Rishon LeZiyyon, Israel; [3]School of Plant Sciences and Food Security, George S. Wise Faculty of Life Sciences, Tel Aviv University, Tel Aviv, Israel; [4]Sagol School of Neuroscience, Tel Aviv University, Tel Aviv, Israel

*For correspondence:
ryaseltzer@gmail.com (RS);
yossiyovel@gmail.com (YY)

[†]These authors contributed equally to this work
[‡]These authors also contributed equally to this work

## eLife Assessment

This study reveals that female moths use ultrasonic sounds emitted by dehydrated plants to guide their oviposition decisions. It highlights sound as an additional sensory modality in host searching, adding an **important** piece to the puzzle of how insects and plants interact. Through **convincing** experimental approaches, the authors provide insights that advance our understanding of plant-insect interactions.

**Abstract** Insects rely on plants' visual, chemical, tactile, and electrical cues when making various decisions. A recent study demonstrated that dehydrated plants emit ultrasonic sounds within the auditory sensitivity range of many moth species. In this study, we sought to determine whether insects also rely on plant acoustic signals when making decisions. We investigated whether female moths rely on ultrasonic clicks which are typically produced by dehydrated plants when deciding where to oviposit. In the absence of an actual plant, the moths indeed preferred to lay their eggs in proximity to acoustic signals which represent dehydrating plants. Tracking the moths' behavior prior to the decision showed that they examined both sides of the arena and gradually spent more time on the acoustic-playback side. Interestingly, when actual plants were added to the arena, the oviposition preference was reversed and the moths preferred silent plants, which is in accordance with their a priori preference for hydrated plants. Deafening the moths eliminated their preference, confirming that the choice was based on hearing. Moreover, the presence of male moths, including their auditory signals, did not affect their oviposition decision, suggesting that the response was specific to plant sound emissions. We reveal evidence for a first acoustic interaction between moths and plants, but as plants emit various sounds, our findings hint at the existence of more currently unknown insect-plant acoustic interactions.

## Introduction

Plant-insect communication has been shown to rely on various modalities, including vision, olfaction, and mechanoreception (*Boppré, 1978*; *Kevan and Lane, 1985*; *Gori, 1989*; *Ne'eman and Nesher, 1995*; *Schiestl, 2010*; *Brito et al., 2015*; *van Dam and Bouwmeester, 2016*). Plant-insect (airborne)

acoustic communication, however, has never been demonstrated. It has long been known that plants vibrate at ultrasonic frequencies due to physiological processes such as cavitation, resulting from changes in their water pressure (*Milburn and Johnson, 1966*; *Tyree and Dixon, 1983*; *Ponomarenko et al., 2014*). Recently, it has also been shown that these ultrasonic sounds produced by a drought-stressed or cut plant are airborne and are probably loud enough to be detected by ultrasound-hearing moths from a distance of a few meters (*Khait et al., 2023*). Moreover, it was shown that these sounds can serve as reliable cues for the condition of the plant, specifically indicating whether a plant is drought-stressed.

Ultrasonic hearing abilities and hearing organs located on different body parts have evolved multiple times independently in several Lepidoptera families. Hearing sensitivity typically falls within the 20–60 kHz range in all groups of moths that have evolved ultrasonic hearing (*Fenton and Fullard, 1979*; *Hoy, 1996*; *Conner, 1999*; *Robert and Göpfert, 2002*; *Moir et al., 2013*; *Göpfert and Hennig, 2016*). Two main hypotheses exist regarding the evolution of these hearing organs. The first suggests that they have evolved for sexual communication, i.e., to detect ultrasonic signals produced by male moths (*Nakano et al., 2009*). The second hypothesis suggests that they have evolved as an anti-predator mechanism to detect echolocation calls produced by bats (*Conner, 1999*; *Greenfield and Weber, 2000*; *Nakano et al., 2015*, but see *Kawahara et al., 2019*). Regardless of why it has evolved, ultrasonic hearing allows moths to detect various additional sounds (*Spangler, 1988*), including plant dehydration sound clicks which have a wide spectrum that overlaps with moths' hearing range and peaks around 50 kHz (*Khait et al., 2023*). We thus hypothesized that herbivore female moths with ultrasonic hearing might exploit ultrasonic plant emissions as cues to infer plant condition and employ this information for oviposition.

The selection of an oviposition site has a significant impact on the fitness of the hatching herbivore larvae and is thus one of the most critical decisions in the life of a female moth (*Lhomme et al., 2018*). In this study, we examined the Egyptian cotton leafworm (*Spodoptera littoralis*; Noctuidae) – a polyphagous herbivore and one of the most significant pests of tomato plants (*Prasad and Bhattacharya, 1975*), which possesses tympanic ears tuned to ultrasonic frequencies (*Tougaard, 1996*; *Skals et al., 2005*; *Anton et al., 2011*). The ears' sensitivity of many moths from the Noctuidae family has been fully characterized, and they typically show a wide range of sensitivity between ~20 and ~60 kHz (*Fullard, 1998*). The full audiogram of the Egyptian cotton leafworm moth has not been documented, but (in accordance with the moths in the Noctuidae family) its hearing has been shown to be most sensitive around 38 kHz, a frequency which is part of the plant's click spectrum (*Tougaard, 1998*). Moreover, the spectra of the clicks of the males of this species (*Figure 1*), which are clearly heard by the females, broadly overlap with plant clicks. We further demonstrated that the moth can hear echolocation calls which are in the range between 40 and 80 kHz, thus demonstrating sensitivity in the plant clicking range (see Methods).

Much research has been conducted to characterize the females' oviposition choice in this species with many factors suggested to be important for their decision-making process. The females have been found to prefer certain species of host plants over others (*Salama et al., 1971*; *Sadek et al., 2010*), to select plants based on their larval experience (*Proffit et al., 2015*) and to choose plants devoid of parasitic larvae, possibly because the presence of such larvae could promote the recruitment of natural enemies (*Sadek et al., 2010*). Studies have also investigated female preferences in response to plant stress signals, particularly olfactory cues. However, there is no clear consensus on the direction of these preferences (e.g. *Chen et al., 2008*; *Showler and Moran, 2003*). Nonetheless, it is widely accepted that females are capable of recognizing and responding to these signals.

In this study, we investigated whether ultrasonic sounds typical of drought-stressed plants influence oviposition decision-making in the Egyptian cotton leafworm moths. Based on their general behavioral preference for non-dry plants (as we validated, see below), we hypothesized that the female moths would be affected by plant ultrasonic signals when making oviposition decisions. Our results support this hypothesis, providing the first evidence for the use of typical plant sounds by insects.

## Results

In each of the following experiments, we placed 10.9±0.17 (mean ± SE) fertile female *S. littoralis* moths in the center of a 100×50×50 $cm^3$ arena divided in the middle, with two choices offered, one

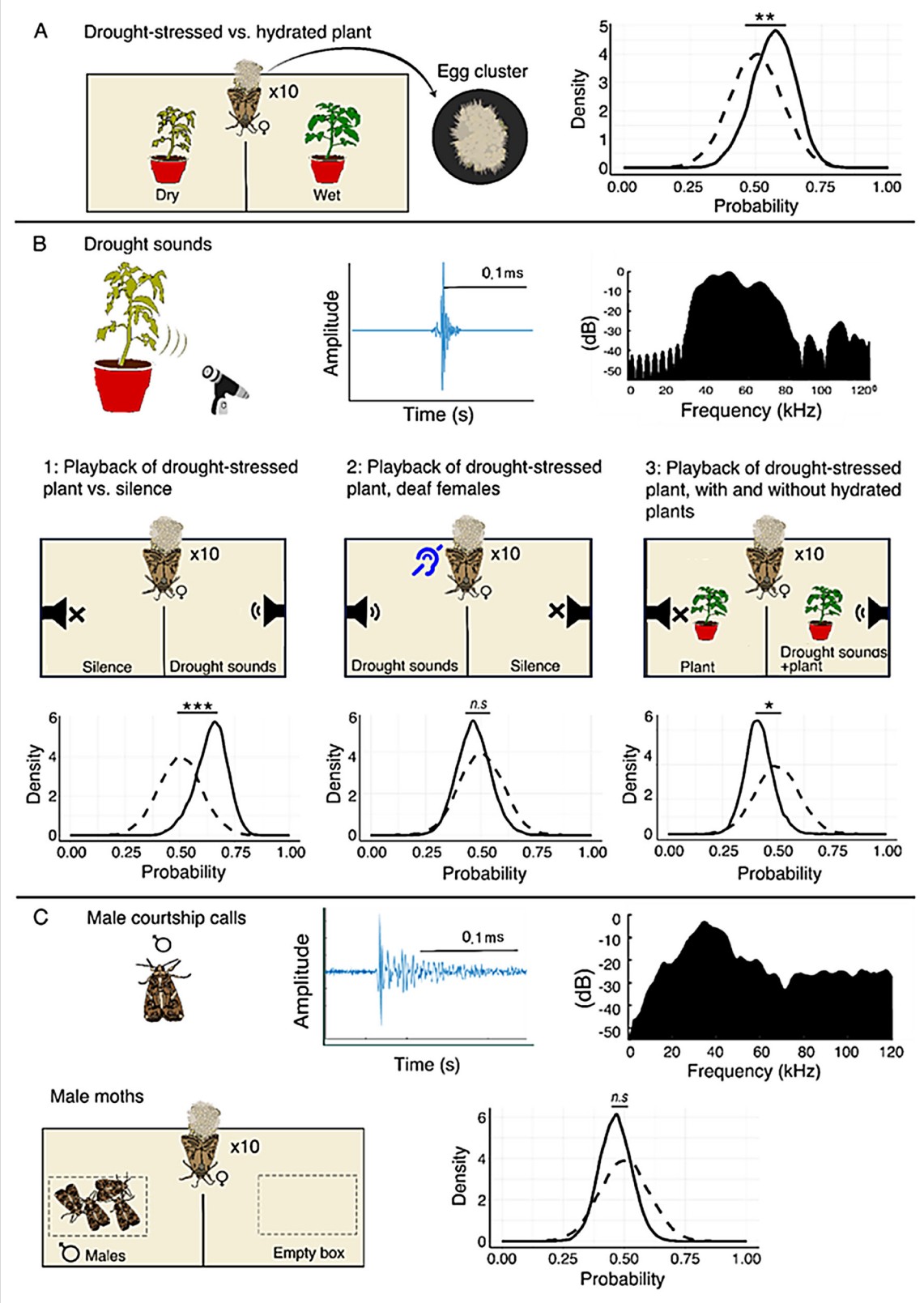

**Figure 1.** The setup and results. In all panels **A–D**, the sound played in the setup is presented in the left section (treatment). Because the number of egg clusters was low (between 0 and 5 clusters), we find that presenting the Bayesian posterior (see Methods) for the probability to lay a cluster is more informative (we present the raw data on *Figure 1—figure supplement 2*). The posterior distribution is depicted by solid lines. The prior distribution (with a mean of 0.5 and a standard deviation of 0.1) is represented by dashed lines. To create these plots, eggs laid on the tested side (where the

*Figure 1 continued on next page*

*Figure 1 continued*

speaker was active or hydrated plant in the initial experiment) are denoted as 1, while those on the opposite side are marked as 0. These plots thus demonstrate the probability of obtaining a 1 or 0 in each experiment. The middle section shows the two-choice oviposition setup, and the right side shows the results for the following conditions: (**A**) Drought-stressed vs. thriving plant (no playback). (**B1**) Silence vs. drought-stressed plant playback (without a plant). (**B2**) Deaf females in a setup with silence vs. drought-stressed plant playback (without a plant). (**B3**) Silent plant vs. playback of drought-stressed plant. (**C**) A box with male moths vs. an empty box. Tomato and male clicks are presented (time signal and spectrum) in panels **B** and **C**. The horizontal black bar depicts 0.1 ms.

The online version of this article includes the following figure supplement(s) for figure 1:

**Figure supplement 1.** Male Egyptian cotton leaf moths (*S. littoralis*) courtship sequences recorded when we placed males in the arena (spectrogram presented).

**Figure supplement 2.** Raw scatter plot data supporting *Figure 1* (two-choice experiments).

on either side of the arena (a two-alternative forced choice paradigm, see Methods). To assess their choice, we compared the number of egg clusters which the moths had laid on each side.

Each treatment was repeated at least nine times (i.e. with a new set of moths), but the moths in each repetition were observed for several consecutive nights so that the minimum number of egg-laying events per treatment was 17. Each night was considered an independent observation because the moth could make a new decision regarding where to lay her eggs (to account for this repetition, the nights were nested in the statistical model). The treatment and the control sides were

**Table 1.** Summary of experimental conditions, including the number of repetitions, i.e., the number of times that new moths were placed in the arenas and the number of observations (each repetition was observed for approximately three consecutive nights). The total number of egg clusters and the p-values for each experiment are reported. Experiments that were replicated twice appear in two separate lines denoted for combined statistics and by #1 or #2. Experiments and observations that did not produce any egg-laying were excluded from the dataset, and that is why the number of observations is often the same as the number of repetitions.

| Experiment | #Repetitions | #Observations | #Egg clusters | Mean ± SE clusters on the side of the treatment | Mean ± SE clusters on the side of the control | p-Value | Estimates (# of egg clusters) |
|---|---|---|---|---|---|---|---|
| Drought-stressed plants vs. well-hydrated plants | 17 | 17 | 53 | 0.88±1.11 | 2.23±2.68 | 0.01 | 0.93 |
| Playback of a drought-stressed plant vs. silence, combined trials (playback: 60 per minute) | 38 | 45 | 67 | 1.08±0.82 | 0.40±0.65 | 0.00 | 1.00 |
| Playback of a drought-stressed plant vs. silence #1 (playback: 60 per minute) | 11 | 17 | 24 | 1.11±0.69 | 0.29±0.58 | 0.01 | 1.34 |
| Playback of a drought-stressed plant vs. silence #2 (playback: 60 per minute) | 27 | 28 | 43 | 1.07±0.89 | 0.46±0.69 | 0.02 | 0.84 |
| Deafened moths –Playback of a drought-stressed plant vs. silence | 23 | 23 | 39 | 0.70±0.70 | 1.00±1.09 | 0.55 | 0.12 |
| Well-hydrated plants and playback of a drought-stressed plant, combined trials (playback: 60 per minute) | 29 | 39 | 110 | 1.05±0.99 | 1.76±1.64 | 0.01 | –0.52 |
| Well-hydrated plants and playback of a drought-stressed plant #1 (playback: 60 per minute) | 9 | 19 | 44 | 0.78±0.91 | 1.52±1.38 | 0.05 | –0.66 |
| Well-hydrated plants and playback of a drought-stressed plant #2 (playback: 60 per minute) | 20 | 20 | 66 | 1.30±1.03 | 2.00±1.86 | 0.10 | –0.43 |
| Males vs. no-males | 19 | 29 | 48 | 0.72±0.92 | 0.93±1.33 | 0.39 | –0.25 |

alternated between repetitions. To ensure replicability, the main plant-acoustic treatments were run twice with a pause of several months in between (see *Table 1* in the Methods). In these experiments, we used the number of egg clusters, rather than the total number of eggs, as the response variable because each cluster represents a distinct oviposition decision. However, we describe a third experiment below where we evaluated the effect of the plant sounds on egg number (and not cluster number).

First, to examine whether *S. littoralis* females prefer to lay their eggs on drying or fresh tomato plants (without any playback sound, see Experiment 1 in the Methods), we placed them in an arena with one drying and one fresh plant. Female *S. littoralis* demonstrated a strong preference to lay their eggs on fresh plants that were not drought-stressed (*Figure 1A*, 2.2±2.7 vs. 0.9±1.1 egg clusters; mean ± SE; clusters per night respectively, p=0.004, mixed effect generalized linear models (GLMM) with the number of egg clusters as the explained parameter, the treatment as a fixed effect and the number of the arena and the repetition round and night as random effects, see Statistics).

We next examined whether an ultrasonic acoustic stimulus affects moths' oviposition decision-making. To this end, we played drought-stressed sounds (recorded from a real drying tomato plant) on one side of the arena and either placed nothing on the other side or placed a decoy silent resistor to control for electric field sensing (see Experiment 2 in the Methods). Because we aimed to examine the effect of sound only (without other sensory cues such as visual or olfactory), in this condition, there was no plant in the arena, and we placed a small mesh box wrapped with a paper towel in the center of each side to encourage oviposition (the speaker was under the mesh so that the moth could not sense the vibration directly, only through airborne sound waves). Female moths significantly preferred to lay their eggs on the side of the arena in which drying plant sounds were played (contradicting the initial observation that they prefer hydrated plants; *Figure 1A vs. 1B1*).

To make sure that the acoustic signals were the sole influential factor in the moths' decision-making process, we deafened mated female moths (by puncturing the tympanic membrane located at the thoracoabdominal juncture using an entomological needle #2, see Methods section) and repeated the experiment (drought-stressed sounds – no plant in the arena). We placed 9.3±1.8 female moths in an arena and monitored their choice of oviposition sites. In accordance with the acoustic hypothesis, the deafened moths did not show any preference in egg-laying (*Figure 1B2*, 0.70±0.70 vs. 1.0±1.09 egg clusters per night, p=0.55, estimate = 0.12, GLMM).

Notably, this experiment was repeated twice – 6 months apart – and the preference was significant both times (*Figure 1B1*, 1.1±0.8 vs. 0.4±0.7 egg clusters per night for the playback and the silent side, respectively, mean ± SE, p=0.0004, estimate = 1, GLMM as above, see *Table 1* for the results of each session). The average number of egg clusters (1.1 clusters per night) in this condition was lower than in the baseline condition with a plant (2.2 clusters), but this is reasonable when taking into account that there was no plant in the arena. The playback rate was high with 60 drought clicks played per minute. This is higher than the rate reported for a single young plant, but it is feasible when considering a patch of adult plants as we have demonstrated experimentally (see Methods). Moreover, we repeated this experiment in an improved experimental setup with a lower playback rate of 30 per minute and got the same result – see below.

To examine the importance of sound in oviposition decision-making under pseudo-natural conditions, we placed two hydrated tomato plants – one on either side of the arena – and added a speaker playing back drought emissions sounds on one side and a resistor on the other (with the same impedance as the speaker) to control for potential effects of the electric field, or nothing. Interestingly, females showed a significant preference for the silent plant. In this case, the female preference was similar to the initial experiment (without playback) in which the females preferred hydrated plants. The females laid 1.8±1.6 vs. 1.1±1.0 egg clusters per night on the silent and playback sides, respectively. This treatment was also repeated twice over a 12-month period (*Figure 1B3*, estimate = –0.52, p=0.01, GLMM as above, see *Table 1* for the results of each repetition, note that the second repeat was only marginally significant).

To assess whether the moths' response was specific to plant sounds, we conducted an additional test using male moths that were placed on one side of the arena (in a mesh box so females could not interact with). The male moths produced courtship clicks with a similar spectral range to tomato clicks (as we validated, Methods). Females showed no significant preference to lay their eggs near male moths (see *Figure 1—figure supplement 1*, *Figure 1C*, p=0.4, estimate = –0.25, GLMM as above).

To gain further insight into the moths' decision-making process, we repeated Experiment 2 (*Figure 1B*) where drought-stressed sounds were played on one side of the arena without a plant in three additional repetitions (with a total of N=13 females) while videoing and tracking the entire behavior. In these repetitions, eggs were laid only on the playback side of the arena. The continuous tracking showed that most moths (8 of the 13) visited both sides of the arena, crossing sides 4.2±5.7 times (mean ± SD) on average during the night (*Figure 2A*). Moreover, over time, there was a significant increase in the female moths' tendency to spend more time in the playback side (logistic GLMM, p<0.004, *Figure 2B*).

The sound gradient experiment: To control for a few of the experimental parameters from the setup shown in *Figure 1*, we conducted another experiment testing the main effect of plant sounds on oviposition. This experiment replicated the oviposition site preference between the plant stress sound side and the quiet side, but within a different experimental setup (see Sound gradient experiment in the Methods). Namely, in this experiment, we tested a single moth each time, with a lower biologically feasible plant click rate (30 clicks per minute, for experiment regarding natural click rate, see Methods) within a long arena – creating a sound gradient. To this end, we placed a single female moth in a 150-cm-long arena. On one side of the arena (location –75, *Figure 3A*), a speaker played sounds recorded from a drought-stressed tomato plant at a rate of 30 clicks per minute. On the other side of the arena (location +75, *Figure 3A*), there was a silent resistor. A feeder with 60% sugar solution was positioned at the center (location 0, *Figure 3A*). For each egg cluster, we then measured the distance from the center where it was laid and the number of eggs it contained. The results, for both egg and cluster numbers, revealed a clear bimodal distribution with peaks near the feeder and the speaker but not at the silent edge of the arena. Hence, most clusters were laid very close to the feeder or the speaker, while no eggs were laid near the resistor (the closest egg was 21 cm away, *Figure 3B*, both egg and cluster number distributions were significantly different from the expected H0 distribution which was estimated using permutation, Kolmogorov-Smirnov [K-S] test, $p=2.2 \times 10^{-16}$ for the clusters, $p=3.9 \times 10^{-14}$ for the eggs, and see Methods and *Figure 3—figure supplement 2*). To exclude any potential effect of temporal correlations on egg-laying, we have also rerun the statistics when only taking the first night, when the females laid clusters to avoid the desensitization or dependency. This test revealed similar results (D=0.55, $p=2.2 \times 10^{-16}$). This was thus a third independent validation that females prefer to lay eggs near plant playback and that this behavior is seen both when quantifying the individual egg or the cluster level.

As noted, moths prefer to oviposit near stress sounds in a plant-free system (*Figure 1B1*), but their response reverses when stress sounds are played in a system containing plants, leading them to choose oviposition sites near the quiet plant (*Figure 1B3*). In this experimental system, we aimed to test whether this reversal reflects a general preference for plants (even when stressed) over no-plant options. We offered moths a dehydrating plant with added clicking sounds on one edge of the arena and plain soil on the other (*Figure 3—figure supplement 2*). Moths significantly preferred to lay their eggs on the dehydrating clicking plant compared to plain soil (*Figure 3—figure supplement 2*). This experiment conceptually simulates one step prior to *Figure 1B1* – removing the stressed plant while retaining only acoustic signals – suggesting that clicking sounds might be perceived as indicative of plant presence in the absence of multimodal signals.

## Discussion

We reveal first evidence for the use of acoustic information and specifically of sounds typically emitted by plants in insect decision-making. Despite decades of research on plant vibrations, it has only recently been shown that these vibrations can be detected remotely by organisms with ultrasonic hearing ability (*Khait et al., 2023*). Our current results suggest that *S. littoralis* females detect and respond to ultrasonic clicks which are typically emitted by drought-stressed tomato plants and adjust their choice of oviposition accordingly. This finding opens a whole new range of possibilities for animal-plant acoustic interactions.

Moreover, the presence of clicking male moths had no significant effect on the females' oviposition preference, suggesting that female moths can distinguish between different sounds and specifically respond to plant-like sounds. Although the moth's hearing system might be too simple to distinguish among the spectral properties of the different sounds, i.e., male clicks vs. plant sounds (*Nakano et al., 2013*), the temporal patterns of the sequences emitted from these sources are very different. While

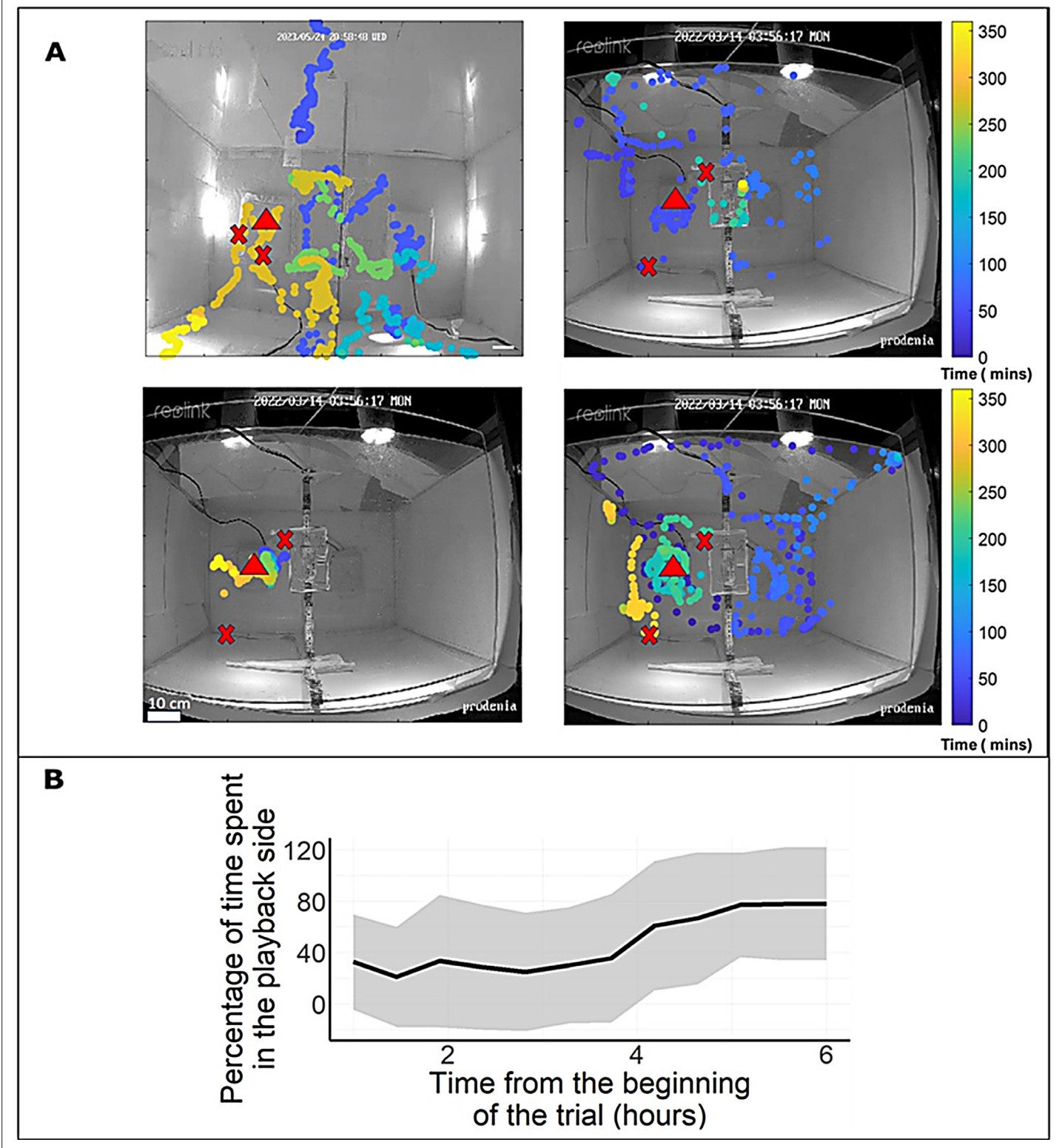

**Figure 2.** Females' movement and decision-making. (**A**) The continuous location over time in the arena (top view) of four individual moths during one trial of the drought sounds vs. silent treatment. Time is represented by color in minutes, with a red triangle indicating the playback side and red X's marking the locations where eggs were laid. Note that we cannot be sure which of the individuals laid the eggs. (**B**) The proportion of time moths spent in the playback side (in bins of 30 min) increased over time.

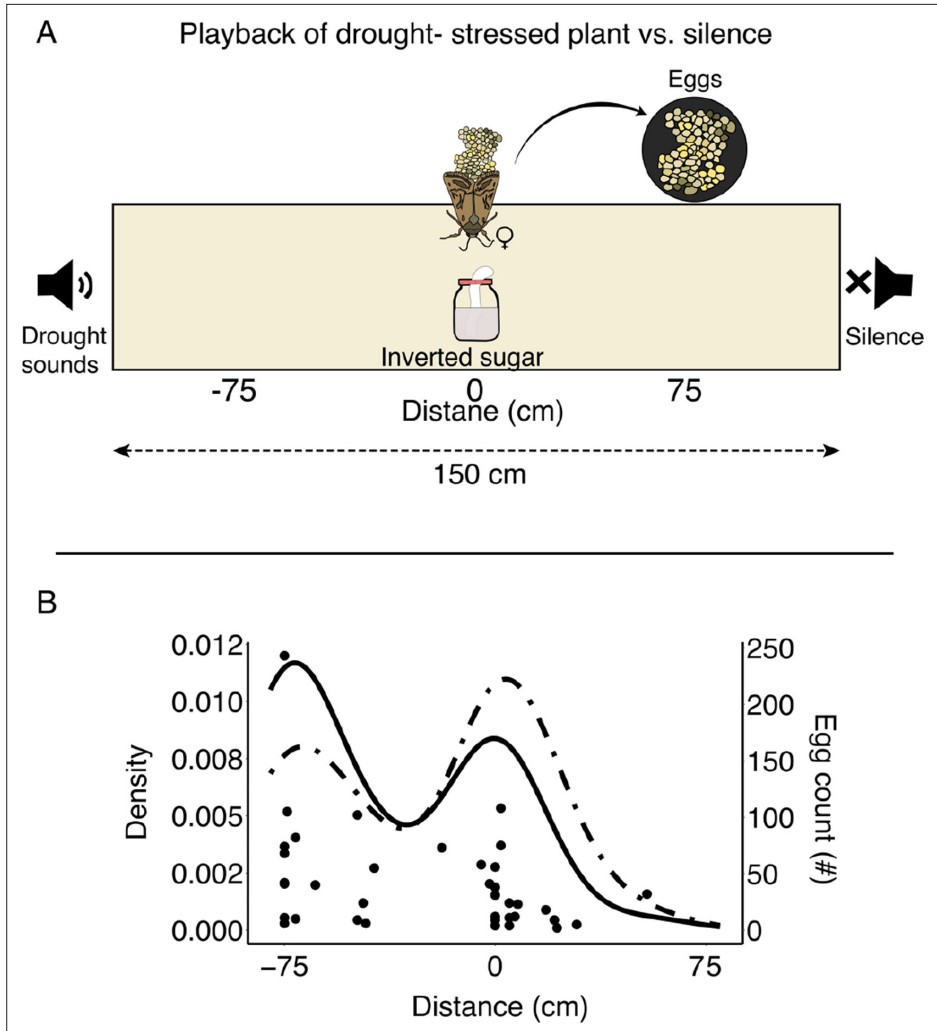

**Figure 3.** Females lay eggs near acoustic playback. (**A**) The long arena creates an acoustic gradient, allowing us to investigate whether female moths prefer to lay their eggs in specific locations based on the sound environment. Additionally, there is sugar water in the center of the arena, which serves as the adult moth's food. (**B**) Egg count density (solid line) and cluster density (dashed line). Both figures display a bimodal distribution, with one peak near the speaker (–75) and another near the feeder (0). The points under the graph depict laid clusters, illustrating the relationship between the number of eggs per cluster and their spatial distribution within the arena.

The online version of this article includes the following figure supplement(s) for figure 3:

**Figure supplement 1.** On the left, comparison between the egg count results (solid line) in the elongated arena and the pseudorandom distribution (dashed line) (Kolmogorov-Smirnov [K-S] test, D=0.3, p=2.2 × 10⁻¹⁶).

**Figure supplement 2.** Sound gradient two-choice test: Drought-stressed playback+ plant vs. quiet control.

male moths emit bursts of several clicks (*Figure 1—figure supplement 1*), plants emit sporadic clicks with no clear temporal order (as used in our playback). Playback of additional sound signals is needed to examine moth specificity.

Although females responded in both treatments when ultrasonic drought-stressed signals were played, they exhibited opposite preferences depending on the presence of a plant. When there was no plant in the arena, the moths showed a strong preference to the playback side, while when plants were present in the arena, the moths switched preference to lay their eggs on the silent side. This latter choice was in accordance with their preference to lay eggs on thriving vs. dry plants, while the first choice (without a plant) was somewhat surprising.

One explanation for this reversal in preference might be the multimodal moth decision-making process. When drought-stressed signals alone (without a plant) were presented to the female moths,

they might have become the only reliable signals for the presence of a plant in the arena, which can explain their strong preference for this side (*Ramaswamy, 1988*; *Sadek, 2011*; *Zhang et al., 2024*). In contrast, when we integrated thriving plants into the arena, the moths' decision-making became multifactorial. Namely, on both sides of the arena, there were visual, texture, and olfactory cues of thriving plants, while the treatment side also exhibited an acoustic signal of a stressed plant. In this setup, the females' oviposition preference was reversed to the side without the acoustic signal. This might suggest that the acoustic signal interpretation is content-dependent, i.e., that the playback of stress sounds in a multifactorial setup became a reliable signal of the physiological state of the plant. Therefore, the females reverted back to their original preference to oviposition on thriving plants.

To further examine this hypothesis, we conducted an additional experiment using the same protocol described for the 'Sound gradient experiment' (see Methods), except that we placed a dehydrated plant (subjected to the stress treatment detailed in Experiment 1) on the side of the speaker that was playing plant sounds. The resulting oviposition pattern closely mirrored those of our earlier studies: when presented with a stressed plant supposedly emitting dehydration sounds, *S. littoralis* females preferred to deposit their eggs on a dehydrated clicking plant rather than on a no-plant control (*Figure 3—figure supplement 2*). These findings imply that a stressed, clicking plant is more attractive for oviposition than an empty substrate, suggesting that clicking might be a cue for the presence of a plant.

Supporting this hypothesis, in the two-choice experiments, the probability of laying eggs at all was significantly higher when a plant was present than in the absence of a plant. Specifically, eggs were laid on 68% vs. 54% of the nights with and without plants, respectively (p=0.009; binomial test comparing Experiments 2 and 3). The number of egg clusters was also higher when a plant was present (see *Figure 1*). We conclude that the moths were more reluctant to lay their eggs when no plant was present.

The preference for the silent plant vs. a plant with stress acoustic playback was not as clear as the preference for the thriving hydrated plants (compare *Figure 1B1 and B3*). There are several potential explanations for this difference. First, moths probably rely on various cues, including olfaction, to detect a drying plant (*Ramaswamy, 1988*; *Sadek, 2011*; *Zhang et al., 2024*). Although the playback allowed us to isolate the specific effect of the acoustic cue, and we tried to select equal plants, we could not control for other cues provided by the plant, and we may have provided the animal with a partial (and likely even contradictory) set of cues. For instance, the plants might have secreted drought-related volatiles and (although watered) might have occasionally emitted sounds spontaneously, reducing the effect of our playback. Indeed, a physiological measurement of plant volatiles suggested that drying plants can be (at least partially) distinguished by the moths.

We further investigated the behavioral mechanism of the female moths as they explored the arena. We quantified the moths' movement during the decision process in the experimental setup with drought-stressed acoustic signals played on one side and with an equal-impedance resistor on the other side. Our findings indicated that their decision process typically included crossing over between the two sides of the arena and spending an increasing amount of time on the (drought-stressed) playback side. This suggests that females explore the available space and ultimately decide based on comparing the two.

Various plant species emit airborne ultrasonic clicks when they are drought-stressed, which can serve as reliable cues for the physiological condition of the plant (*Khait et al., 2023*). Our findings demonstrate that moths with auditory abilities use these clicks when choosing a site for oviposition. We hypothesize that some other species of insects might also exploit these acoustic cues to their advantage in different contexts. Pollinating insects, for example, might use drought-related sounds when choosing where to forage. Some insects might even be able to distinguish between clicks produced by different plants or under different conditions, such as drying plants vs. plants under a pathogen attack.

Plant clicks are ultrasonic and thus very different from most other outdoors sounds (such as wind sounds, as we also show in *Khait et al., 2023*). Moreover, because the clicks are ultrasonic and not very intense, they can only be picked up by the moths from a short distance (~1.5 m), which allows the moths to localize them in space.

The sounds emitted by drought-stressed plants are probably a cue rather than a signal, i.e., they did not evolve to convey information to insects. The interaction that we have demonstrated in this

study therefore cannot be considered 'communication' according to the conservative definition of the term, which relies on signals that have evolved to convey a specific message (*Searcy and Nowicki, 2005*; *Skyrms, 2010*). However, it is possible that some plants have evolved an ability to amplify their emissions or modify their spectral content to facilitate desirable interactions with animals and perhaps even with other plants (*Veits et al., 2019*). One exciting possibility would be that plants signal an insect attack by amplifying click intensity to recruit potential predators of the attacking insects, such as predatory insects, rodents, or bats. Such amplification could be achieved by various morphological modifications. Insects, on the other hand, might have evolved behavioral strategies to move near plants and pick up these weak acoustic signals. In conclusion, our study shows that moths are able to detect and respond to acoustic signals emitted by plants. This discovery suggests the existence of a third type of acoustic signal that moths utilize, in addition to those produced by bat echolocation and moth courtship clicks, raising new questions about the evolution of moth hearing. We predict that future studies will uncover more examples of acoustic communication between plants and animals.

## Methods

Experimental setup: We collected pupae of *S. littoralis* that were reared under controlled breeding conditions (reared on castor bean leaves, 25 ± 1°C, 40% relative humidity with a 12–12 hr light–dark cycle). Newly emerged female and male moths were placed together until egg-laying was detected (approximately 2 days). Then, we transferred the females to an experimental arena. Each arena was 100×50×50 cm³ in size, divided in the middle by a plastic partition half the height of the arena (*Figure 1A*). On the partition, we placed a closed test tube with cotton wool containing 60% inverted sugar solution for ad libitum feeding throughout the experiment. Experiment 1 (see below) was performed in a greenhouse (2.5×4.5×3.5 m³) to simulate optimal conditions for plant development. The experiments involving acoustic signals (see below Experiments 2, 3, 4, 5, and 6) were performed in an acoustically shielded room (2.5×4×2.5 m³) to prevent acoustic interference. Each of the following treatments was performed simultaneously in up to four arenas. Moths could choose between the treatments presented on each side of the arena (see below), and oviposition was monitored daily for 3 days by counting the number of egg clusters. At the end of each night, we cleaned the arena of counted egg clusters using a cloth with ethanol, so that on the subsequent night, we would not expect there to be evidence of previous oviposition. We repeated the experiments under the same conditions until acquiring at least nine nights with egg-laying observations (eggs were not always laid, which is not surprising given the artificial conditions in the acoustic room used for these experiments). We refer to the cluster and not to the individual egg as the moth's decision unit, because each cluster requires a decision about the location of oviposition, whereas the number of eggs could be affected by the general condition of the female or by external interference. Indeed, there was much variation in the number of eggs per cluster – 68±134 eggs (mean ± SE). However, to determine whether counting eggs would have altered our results, we conducted an experiment comparing cluster counts to individual egg counts (Experiment 6). For experiments with actual plants, a young tomato plant (*Solanum lycopersicum*) in a small pot was used in all experiments. All the treatments are illustrated in *Figure 1A–C*. The number of repetitions of each treatment is noted in *Table 1*, and data is presented in *Source data 1*. To maintain moths' vitality through the experiment, we have placed on the starting point (central platform) a closed test tube with cotton wool containing a 60% sugar solution for ad libitum feeding.

1. Drought-stressed vs. well-hydrated plants: We placed a single-stem tomato plant, 10 cm high, on either side of the arena. The plant on one side was drought-stressed (3 days without watering), and the other was thriving and well-hydrated. Moths could lay eggs on either plant (*Figure 1A*).

2. Playback of a drought-stressed plant vs. silence (without plants): Each side contained an oviposition box (10×15×5 cm³ made of 0.5×0.5 cm² mesh), covered with a paper towel. A speaker playing sounds recorded from a drought-stressed tomato plant (*Khait et al., 2023*) was placed under one of the two oviposition boxes (on one side of the arena). The speaker played drought sounds at the same intensity measured for real plants at a rate of 1 click per minute, with a stochastic 10% error in the intervals between clicks (see below for details on assessing intensity and playback rate). The oviposition box on the other side either had a resistor similar to the speaker in shape and identical in impedance to control for potential effects of the electric field created by the speaker (though we did not account for a magnetic field produced by

the speaker, which might as well affect the choice) or no resistor (we did not find significant differences between the two silent controls, GLMM, p=0.58). The experiment was performed twice to strengthen the confidence in its results: the first trial was performed during August and September 2021 and the second during February to May 2022 (a pool of both trials and controls – with and without resistors – is presented in *Figure 1B1*). We also repeated this experiment a third time with a lower emission rate of 2 clicks per minute (*Table 1*).

3. Deaf females in a setup with silence vs. drought-stressed plant playback (without a plant): We deafened mated females by puncturing their tympanic membrane and placed them in an arena to assess their response to drought-stressed sounds, compared to a silent control (as described in Experiment 2). Deafening surgical procedure: We performed a surgical procedure on female moths to deafen them. The procedure involved puncturing the tympanic membrane located at the thoracoabdominal juncture using an entomological needle #2. The female moths recovered from the procedure within 2 min and were able to fly normally. We tested a sample of these females in a standard rearing box and found that they were able to lay eggs normally. To confirm that the surgery had successfully deafened the females, we conducted an inspection by playing a bat playback (the same as described below). We deafened a group of 20 moths and compared their reactions to a control group of 25 non-deafened moths. During the experiment, the moths were released in a dark acoustically isolated room ($5.5 \times 4.5 \times 2.5$ m³) with acoustic foam on the walls and ceiling and a single light source (12 W mercury vapor bulb peaked at 1650 lux), and while they were in flight around the light source, we emitted the sound. In the control group, five moths exhibited a response (such as falling or a significant change in direction) upon hearing the sound (scored by a naïve viewer who did not know whether the moths were treated). In contrast, none of the deafened moths displayed any reaction to the clicking stimulus (Q=4.5, p=0.03, chi-square test).

4. Well-hydrated plants with and without playback of drought-stressed plant sound: There was an oviposition box on each side of the arena. One side played drought-stressed sounds while the other remained silent, with either a resistor or no sound (same as Experiment 2). Additionally, a thriving, healthy tomato plant was placed on each oviposition box. This experiment was performed twice, 12 months apart, to strengthen the confidence of its results (a pool of both trials and controls [with and without resistors] is presented in *Figure 1B3*). To determine the specificity of the response to plant sounds, two additional controls were performed.

5. Male moths: Five males were enclosed under the oviposition box with sugar water to maintain them. The control box had only sugar water without any moths (*Figure 1C*). We validated that males in this condition produced clicks by recording the sounds emitted by the five male moths enclosed overnight in an acoustically isolated container, which showed that the males frequently clicked. The test was repeated five times and clicks were always emitted by the males (*Figure 1—figure supplement 1*).

## Playback

Drought sounds were recorded using an Hm16 Avisoft microphone and an HM116 Avisoft A/D from a distance of 10 cm in an isolated container with walls covered with acoustic foam (*Khait et al., 2023*). These recordings revealed emission intensities of at least 60 dB SPL (Re 20 µPa) at a distance of 10 cm. The sounds were played using a Vifa speaker connected to an Avisoft D/A converter (Player 116).

We ensured that playback sound intensity was similar to that measured in real plants on the playback side of the arena (i.e. ~60 dB SPL at a distance of 10 cm) and that sound level on the control side was below the detection range of our system, i.e., below 30 dB SPL at 10 cm. We performed four calibration measurements using a calibrated GRAS 40DP microphone during the period of the experiments to validate that sound levels had not changed over time. Using the GRAS calibrated microphone, we also validated that the average sound intensity of the male moth sequences was the same as that of the playback plant sounds.

Validating the playback rate: The drying plant sounds in the box arenas (Experiments 2–4, *Figure 1B*) were played back at a rate of 1 click per second (1 Hz), with up to 10% error in the intervals (caused by the computer controlling the system). This frequency is substantially higher than that found for a single young tomato plant (*Khait et al., 2023*). However, the rate that we played (60 clicks per minute) is ecologically relevant when considering a patch of tomato (or other) plants. To validate this, we aggregated 45 tomato seedlings in a planting tray ($30 \times 30$ cm²) and placed the tray in an empty greenhouse. The plants were not watered for 3 days, and we recorded sound continuously for 50 hr (using the same Hm116 microphone setup noted above). When placing the microphone ~20 cm

above the tray – as a flying moth would do, we measured a maximum click rate of 20 clicks per minute (i.e. 0.33 Hz). This is threefold slower than the rate we used, but very similar to the rate that we used in the gradient experiment (see below). Moreover, when taking into account the moth's detection range for this emission intensity, which is likely ~1.5 m at least (*Khait et al., 2023*), a female moth could be exposed to a rate over threefold higher (i.e. higher than 1 Hz) in a patch of drying plants (which would contain more than 100 seedlings in a typical bush of agricultural or wild hosts typical of this species). Notably, every plant that we examined was found to emit similar ultrasonic clicks when dehydrating (*Khait et al., 2023*), so this behavior could be relevant to other plants, many of which grow as dense bushes.

### Sound gradient experiment

We used elongated arenas (150×20×5 cm$^3$, *Figure 3A*). In the center of the arena (location 0), we placed a closed test tube with cotton wool containing a 60% sugar solution for ad libitum feeding (to maintain moths' vitality). To facilitate accurate measurement of egg distances from the speaker (location –75), we printed a ruler and placed it along the bottom of the arena. Each moth was placed at the center of the arena at the beginning of the experiment. On the next morning, we recorded the locations of the egg clusters and counted the number of eggs in each cluster using a stereoscopic microscope, or a magnifying glass if the eggs were not laid on the ruler. Each female remained in the arena during the days starting 3 days after emerging from the pupa and mating and until it died. After each night, we switched the locations of the speaker and the resistor within the arena. We measured a 30 dB SPL difference in intensity between the side of the speaker and the side of the resistor. The clicks were emitted at a frequency of 0.5 clicks per second (30 per minute).

### Tracking the females' decision-making process

In order to investigate how moths survey the experimental arena and subsequently engage in a decision-making process, we conducted two additional trials in which we continuously recorded the movement of the moths throughout the night. In each trial, we placed four female moths on a platform in the middle of the arena, in which a speaker played drought-stressed plant sounds on one side, while on the other, control side, we placed a silent resistor (as in treatment 3 above). We exchanged sides between trials and tracked the moths for 6 hr using an IR camera (Reolink RLC-511-5MP camera) placed above the arena. We then documented the position of each moth at 12 s intervals using the DLTdv 8 software (*Hedrick, 2008*). Each individual was recognized according to its proximity to the last tracking point in order to reconstruct its full movement. We quantified how many times each individual crossed the center of the arena (the platform in the center was divided in the middle) and the proportion of time it spent in each side.

### Statistics

GLMMs were used in MATLAB to examine the females' choice of oviposition. Random effects were set as intercepts. The number of clusters was set as the explained variable. The treatment, i.e., playback or control, and the number of female moths in the arena, was set as a fixed effect. The number of the arena, the month in which the experiment was performed, the number of repetitions, and the night of the repetition were considered as random effects. Because we were analyzing counts (number of clusters), the model was run using a Poisson distribution. In the experiments in which we ran two repetitions of the same experiment, we added the session as another fixed parameter and we also ran the statistics separately for each session.

To deepen our understanding of the trends observed in the experiments, we implemented Bayesian model fittings for each choice-based experiment. In this analysis, 'oviposition choices' were considered as distinct decisions. A value of 1 was assigned when the egg cluster was located on the side with the active speaker (or on the hydrated plant in the initial experiment) and a value of 0 was assigned for oviposition on the opposite side. We employed a Gaussian model, incorporating the number of females in each experiment as a random effect, with a prior mean of 0.5 and a standard deviation of 0.1. For each experiment, we sampled our data 16,000 times to calculate the posterior distribution from these samples. We used a binomial GLMM to determine the effect of the treatment on the moths' decision-making. To achieve this, the proportion of time spent in each side of the arena was set as an explained variable, the playing side as a fixed effect, with the trial and the individual moth

as random effects. To study the effect of time on the movement of the moths, we used logistic GLMM in which the accumulated amount of time spent on the sound-playing side was set as an explained variable, the time as a fixed effect, and the trial and the individual moths were set as random effects.

To compare the distribution of eggs in the elongated arena to a random distribution, we generated an H0 distribution by randomly shuffling the locations of the speaker and resistor for each laid egg. This distribution was then compared to our actual egg count distribution using the K-S test (*Figure 3—figure supplement 1*).

## Acknowledgements

We thank Hadas Marcus for linguistic editing, Tal Erez for providing moths, Adi Segal, Einav Balachsan, Michael Martynenko, Shira Fraenkel, Yuval Lustig, and Morad Garam for experimental setup, Nitzan Shahar for statistical consulting, Aya Goldshtein for reviewing the manuscript and statistical consulting, Mor Taub for graphic editing, Yonatan Nathan, and Inon Scharf for reviewing the manuscript. This work was supported by the European Research Council (ERC) (PLANT BIOACOUSTICS, project 101098318).

## Additional information

### Funding

| Funder | Grant reference number | Author |
|---|---|---|
| European Research Council | 10.3030/101001993 | Lilach Hadany |
| European Research Council | 10.3030/101098318 | Lilach Hadany |

The funders had no role in study design, data collection and interpretation, or the decision to submit the work for publication.

### Author contributions

Rya Seltzer, Guy Zer Eshel, Conceptualization, Data curation, Software, Formal analysis, Validation, Investigation, Visualization, Methodology, Writing – original draft, Project administration, Writing – review and editing; Omer Yinon, Ahmed Afani, Ofri Eitan, Sabina Matveev, Galina Levedev, Michael Davidovitz, Tal Ben Tov, Gayl Sharabi, Yuval Shapira, Neta Shvil, Maya Harari Gibli, Ireen Atallah, Sahar Hadad, Data curation; Dana Ment, Resources; Lilach Hadany, Conceptualization, Supervision, Investigation, Writing – review and editing; Yossi Yovel, Conceptualization, Supervision, Investigation, Methodology, Writing – review and editing

### Author ORCIDs

Rya Seltzer ⓘ https://orcid.org/0000-0002-3041-4648
Guy Zer Eshel ⓘ https://orcid.org/0009-0007-6828-1290
Lilach Hadany ⓘ https://orcid.org/0000-0002-1642-5308
Yossi Yovel ⓘ https://orcid.org/0000-0001-5429-9245

Reviewer #1 (Public review): https://doi.org/10.7554/eLife.104700.3.sa1
Reviewer #2 (Public review): https://doi.org/10.7554/eLife.104700.3.sa2
Author response https://doi.org/10.7554/eLife.104700.3.sa3

## Additional files

### Supplementary files
MDAR checklist

Source data 1. Raw data tables and analysis code that constitute the primary sources for all analyses in this study.

## Data availability

All tables and codes are available via: https://data.mendeley.com/datasets/yg7ms8rn37/1.

The following dataset was generated:

| Author(s) | Year | Dataset title | Dataset URL | Database and Identifier |
|---|---|---|---|---|
| Zer G | 2025 | Female Moths Incorporate Plant Acoustic Emissions into Their Oviposition Decision-Making Process | https://doi.org/10.17632/yg7ms8rn37.1 | Mendeley Data, 10.17632/yg7ms8rn37.1 |

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
