## [Editor Report · eLife Assessment]

This study reveals that female moths use ultrasonic sounds emitted by dehydrated plants to guide their oviposition decisions. It highlights sound as an additional sensory modality in host searching, adding an **important** piece to the puzzle of how insects and plants interact. Through **convincing** experimental approaches, the authors provide insights that advance our understanding of plant-insect interactions.

---

## [Referee Report · Reviewer #1 (Public review)]

Thank you for your thorough reply to my review and for conducting the additional experiments. All points from my previous review have been addressed.

---

## [Referee Report · Reviewer #2 (Public review)]

This paper presents interesting and fresh approach as it investigates whether female moths utilize plant-emitted ultrasounds, particularly those associated with dehydration stress, in their egg-laying decision-making process. It provides the first empirical evidence suggesting that acoustic information may contribute to insect-plant interactions.

The revised version is significantly strengthened by the addition of supplementary data and improved explanations. The authors present robust results across multiple experiments, enhancing the credibility of their conclusions.

Female moths showed a preference for moist, fresh plants over dehydrated ones in experiments using actual plants. Additionally, when both plants were fresh but ultrasonic sounds specific to dehydrated plants were presented from one side, the moths chose the silent plant. However, in experiments without plants, contrary to the hypothesis derived from the above results, the moths preferred to oviposit near ultrasonic playback mimicking the sounds of dehydrated plants.

These results clearly indicate that moths can perceive plant presence through sound. The findings also highlight the need for future investigation into the multi-modal nature of moth decision-making, as acoustic cues alone may not fully explain the behavioral choices observed across different contexts.

Overall, the results are intriguing, and I think the experiments are very well designed. The authors successfully demonstrate that plant-derived acoustic signals influence oviposition behavior in female moths, thereby achieving the study's objectives. The experimental design and analysis protocols are reproducible and well suited for adaptation to other species.

---

## [Author Response]

The following is the authors’ response to the original reviews.

**Reviewer #1 (Public review):**
(1) The authors demonstrate that female Spodoptera littoralis moths prefer to oviposit on wellwatered tomato plants and avoid drought-stressed plants. The study then recorded the sounds produced by drought-stressed plants and found that they produce 30 ultrasonic clicks per minute. Thereafter, the authors tested the response of female S. littoralis moths to clicks with a frequency of 60 clicks per minute in an arena with and without plants and in an arena setting with two healthy plants of which one was associated with 60 clicks per minute. These experiments revealed that in the absence of a plant, the moths preferred to lay eggs on the side of the area in which the clicks could be heard, while in the presence of a plant the S. littoralis females preferred to oviposit on the plant where the clicks were not audible. In addition, the authors also tested the response of S. littoralis females in which the tympanic membrane had been pierced making the moths unable to detect the click sounds. As hypothesised, these females placed their eggs equally on both sites of the area.Finally, the authors explored whether the female oviposition choice might be influenced by the courtship calls of S. littoralis males which emit clicks in a range similar to a drought-stressed tomato plant. However, no effect was found of the clicks from ten males on the oviposition behaviour of the female moths, indicating that the females can distinguish between the two types of clicks. Besides these different experiments, the authors also investigated the distribution of egg clusters within a longer arena without a plant, but with a sugar-water feeder. Here it was found that the egg clusters were mostly aggregated around the feeder and the speaker producing 60 clicks per minute. Lastly, video tracking was used to observe the behaviour of the area without a plant, which demonstratedthat the moths gradually spent more time at the arena side with the click sounds.

We thank the reviewers for their helpful comments. We agree with the summary, but would like to note that in the control experiment (Figure 2) we used a click rate of 30 clicks per minute—a design choice driven by the editor’s feedback. We have clarified this and, to further probe the system’s dynamics, added a second experiment employing the same click rate (30 clicks per minute) with a dehydrated plant (see details below). In both experiments, females again showed a clear tendency to oviposit nearer the speaker; these findings are described in the updated manuscript.

(2) The study addresses a very interesting question by asking whether female moths incorporate plant acoustic signals into their oviposition choice, unfortunately, I find it very difficult to judge how big the influence of the sound on the female choice really is as the manuscript does not provide any graphs showing the real numbers of eggs laid on the different plants, but instead only provides graphs with the Bayesian model fittings for each of the experiments. In addition, the numbers given in the text seem to be relatively similar with large variations e.g. Figure 1B3: 1.8 {plus minus} 1.6 vs. 1.1 {plus minus} 1.0. Furthermore, the authors do not provide access to any of the raw data or scripts of this study, which also makes it difficult to assess the potential impact of this study. Hence, I would very much like to encourage the authors to provide figures showing the measured values as boxplots including the individual data points, especially in Figure 1, and to provide access to all the raw data underlying the figures.

We acknowledge that there are researchers who favor Bayesian graphical representation versus raw data visualization. Therefore, we have added chartplots of the raw data from Figure 1 in the supplementary section. We are aware of the duplication in presentation and apologize for this redundancy.

Regarding the variance and means we obtained in our experiment, we have analyzed all raw data using the statistical model presented, and if statistical significance was found despite a particular mean difference or variance, this is meaningful from a biological perspective. One can certainly discuss whether this difference has biological importance, but it should be remembered that in this experimental system, we are trying to isolate the acoustic signal from a complex system that includes multiple signals. Therefore, at no point we’ve suggested that this is a standalone factor, but rather proposed it as an informative and significant component.

In addition to the experiments described above, we conducted an experiment in which we counted both eggs and clusters. The results indicate that cluster counts are a reliable proxy for reproductive investment at a given location. In this experiment, we present cluster numbers alongside egg counts (Figure 2).

Furthermore, we apologize for the technical error that prevented our uploaded data files from reaching the reviewers. We have also uploaded updated data and code.

(3) Regarding the analysis of the results, I am also not entirely convinced that each night can be taken as an independent egg-laying event, as the amount of eggs and the place were the eggs are laid by a female moth surely depends on the previous oviposition events. While I must admit that I am not a statistician, I would suggest, from a biological point of view, that each group of moths should be treated as a replicate and not each night. I would therefore also suggest to rather analyse the sum of eggs laid over the different consecutive nights than taking the eggs laid in each night as an independent data point.

We thank the reviewer for this question. This is a valid and point that we will address in three aspects:

First, regarding our statistical approach, we used a model that takes into account the sequence of nights and examines whether there is an effect of the order of nights, i.e., we used GLMMs, with the night nested within the repetition. This is equivalent to addressing this as a repeated measure and is, to our best knowledge, the common way to treat such data.

Second, following the reviewer's comment, we also reran the statistics of the third experiment (i.e., “sound gradient experiments”, Figure 2 and Supplementary figure 4) when only taking the first night when the female/s laid eggs to avoid the concern of dependency. This analysis revealed the same result – i.e., a significant preference for the sound stimulus. We have now updated our methods and results section to clarify this point.

Third, an important detail that may not have been clearly specified in the methods: at the end of each night, we cleaned the arena of counted egg clusters using a cloth with ethanol, so that on the subsequent night, we would not expect there to be evidence of previous oviposition but thus would not exclude some sort of physiological or cognitive memories. We have now updated our methods section to clarify this important procedural point.

(4) Furthermore, it did not become entirely clear to me why a click frequency of 60 clicks per minute was used for most experiments, while the plants only produce clicks at a range of 30 clicks per minute. Independent of the ecological relevance of these sound signals, it would be nice if the authors could provide a reason for using this frequency range. Besides this, I was also wondering about the argument that groups of plants might still produce clicks in the range of 60 clicks per minute and that the authors' tests might therefore still be reasonable. I would agree with this, but only in the case that a group of plants with these sounds would be tested. Offering the choice between two single plants while providing the sound from a group of plants is in my view not the most ecologically reasonable choice. It would be great if the authors could modify the argument in the discussion section accordingly and further explore the relevance of different frequencies and dBlevels.

This is an excellent point. We originally increased the click rate generate a strong signal. However, it was important for us to verify that there was ecological relevance in the stimulus we implemented in the system. For this purpose, we recorded a group of dehydrated plants at a distance of ~20cm and we measured a click rate of 20 clicks per minute (i.e., 0.33 Hz) (see Methods section). Therefore, as mentioned at the beginning of this letter, in the additional experiment described in Figure 2, we reduced the click frequency to 30 clicks per minute, and at this lower rate, the effect was maintained. Increasing plant density would probably lead to a higher rate of 30 clicks per minute.

(5) Finally, I was wondering how transferable the findings are towards insects and Lepidopterans in general. Not all insects possess a tympanic organ and might therefore not be able to detect the plant clicks that were recorded. Moreover, I would imagine that generalist herbivorous like Spodoptera might be more inclined to use these clicks than specialists, which very much rely on certain chemical cues to find their host plants. It would be great if the authors would point more to the fact that your study only investigated a single moth species and that the results might therefore only hold true for S. littoralis and closely related species, but not necessary for other moth species such as Sphingidae or even butterflies.

Good point. Our research uses a specific model system of one moth species and one plant species in a particular plant-insect interaction where females select host plants for their offspring. As with any model-based research that attempts to draw broader conclusions, we've taken care to distinguish between our direct findings and potential wider implications. We believe our system may represent mechanisms relevant to a wider group of herbivorous insects with hearing capabilities, particularly considering that several moth families and other insect orders can detect ultrasound. However, additional research examining more moth and plant species is necessary to determine how broadly applicable these findings are. We have made these clarifications in the text.

**Reviewer #2 (Public review):**
(6) The results are intriguing, and I think the experiments are very well designed. However, if female moths use the sounds emitted by dehydrated plants as cues to decide where to oviposit, the hypothesis would predict that they would avoid such sounds. The discussion mentions the possibility of a multi-modal moth decision-making process to explain these contradictory results, and I also believe this is a strong possibility. However, since this remains speculative, careful consideration is needed regarding how to interpret the findings based solely on the direct results presented in the results section.

Thank you for this insightful observation. We agree that the apparent attraction of females to dehydrated-plant sounds contradicts our initial prediction. Having observed this pattern consistently across multiple setups, we have now added a targeted choice experiment to the revised manuscript: here female moths were offered a choice between dehydrated plants broadcasting their natural ultrasonic emissions and a control. These results—detailed in the Discussion and presented in full in the Supplementary Materials (Supplementary Figure 4)—show that when only a dehydrated plant is available, moths would prefer it for oviposition, supporting our hypothesis that in the absence of a real plant, the plant’s sounds might represent a plant..

(7) Additionally, the final results describing differences in olfactory responses to drying and hydrated plants are included, but the corresponding figures are placed in the supplementary materials. Given this, I would suggest reconsidering how to best present the hypotheses and clarify the overarching message of the results. This might involve reordering the results or re-evaluating which data should appear in the main text versus the supplementary materials

Thank you for this suggestion. We have reorganized the manuscript and removed the olfactory response data from the current version to maintain a focused narrative on acoustic cues. We agree that a detailed investigation of multimodal interactions deserves a separate study, which we plan to pursue in future work.

(8) There were also areas where more detailed explanations of the experimental methods would be beneficial.

Thank you for highlighting this point. We have expanded and clarified the Methods section to provide comprehensive detail on our experimental procedures.

**Reviewer #1 (Recommendations for the authors):**
(9) Line 1: Please include the name of the species you tested also in the title as your results might not hold true for all moth species.

We do not fully agree with this comment. Please see comment 5.

(10) Line 19-20: Please rephrase the sentence so that it becomes clear that the "dehydration stress" refers to the plant and not to the moths.

Thank you for the suggestion; we have clarified the text accordingly

(11) Line 31: Male moths might provide many different signals to the females, maybe better "male sound signals" or similar.

Thank you for the suggestion; we have clarified the text accordingly.

(12) Line 52-53: Maybe mention here that not all moth species have evolved these abilities.

Thank you for the suggestion; we have clarified the text accordingly.

(13) Line 77: add a space after 38.

Thank you for the suggestion; we have clarified the text accordingly.

(14) Line 88: Maybe change "secondary predators" to "natural enemies".

Thank you for the suggestion; we have clarified the text accordingly.

(15) Line 134: Why is "notably" in italics? I would suggest using normal spelling/formatting rules here.

Thank you for the suggestion; we have clarified the text accordingly.

(16) Line 140-144: If you did perform the experiment also with the more ecological relevant playback rate, why not present these findings as your main results and use the data with the higher playback frequency as additional support?

Thank you for this suggestion. We agree that the ecologically relevant playback data are important; as described in detail at the beginning of this letter and also in comment 4, however, to preserve a clear and cohesive narrative, we have maintained the original ordering of this section. Nevertheless, the various experiments conducted in Figure 1 differ in several components from Figure 2 and the work that examined sounds in plant groups in the appendices. Therefore, we find it more appropriate to use them as supporting evidence for the main findings rather than creating a comparison between different experimental systems. For this reason, we chose to keep them as a separate description in "The ecological playback findings (Lines 140–144) remain fully described in the Results and serve to reinforce the main observations without interrupting the manuscript's flow.

(17) Line 146: Please explain already here how you deafened the moths.

Thank you for the suggestion; we have clarified the text accordingly.

(18) Line 181: should it be "male moths' " ?

Thank you for the suggestion; we have clarified the text accordingly.

(19) Line 215: Why is "without a plant" in italics? I would suggest using normal spelling/formatting rules here.

Thank you for the suggestion; we have clarified the text accordingly.

(20) Line 234: I do not understand why this type of statistic was used to analyse the electroantennogram (EAG) results. Would a rather simple Student's t-test or a Wilcon rank sum test not have been sufficient? I would also like to caution you not to overinterpret the data derived from the EAG, as you combined the entire headspace into one mixture it is no longer possible to derive information on the different volatiles in the blends. The differences you observe might therefore mostly be due to the amount of emitted volatiles.

We have reorganized the manuscript and removed the olfactory response data from the current version to maintain a focused narrative on acoustic cues (See comment 7).

(21) Line 268: It might be nice to add an additional reference here referring to the multimodal oviposition behaviour of the moths.

Thank you for the suggestion; we have clarified the text accordingly.

(22) Line 284: If possible, please add another reference here referring to the different cues used by moths during oviposition.

Thank you for the suggestion; we have clarified the text accordingly.

(23) Line 336: What do you mean by "closed together"?

Thank you for the suggestion; we have clarified the text accordingly.

(24) Line 434-436: Please see my overall comments. I do not think that you can call it ecologically relevant if the signal emitted by multiple plants is played in the context of just a single plant.

Please see comments 1 and 4.

(25) Line 496: Please change "stats" to statistics.

Thank you for the suggestion; we have clarified the text accordingly.

(26) Line 522-524: I am not sure whether simply listing their names does give full credit to the work these people did for your study. Maybe also explain how they contributed to your work.

Thank you for the suggestion; we have clarified the text accordingly.

**Reviewer #2 (Recommendations for the authors):**
(27) L54 20-60kHz  20Hz-60kHz or 20kHz - 60kHz?

OK. We have replaced it.

(28) L124 Are the results for the condition where nothing was placed and the condition where a decoy silent resistor was placed combined in the analysis? If so, were there no significant differences between the two conditions? Comparing these with a condition presenting band-limited noise in the same frequency range as the drought-stressed sounds might also have been an effective approach to further isolate the specific role of the ultrasonic emissions.

We have used both conditions due to technical constrains and pooled them tougher for analysis— statistical tests confirmed no significant differences between them—and this clarification has now been added to the Methods section including the results of the statistical test.

(29) L125 (Fig. 1A), see Exp. 1 in the Methods. -> (Fig.1B. See Exp.1 in the Methods).

Thank you for the suggestion; we have clarified the text accordingly.

(30) L132 "The opposite choice to what was seen in the initial experiment (Fig.1B)"

Thank you for the suggestion; we have clarified the text accordingly.

(31) L137-143 If you are writing about results, why not describe them with figures and statistics? The current description reads like a discussion.

These findings were not among our primary research questions; however, we believe that including them in the Results section underscores the experimental differences. In our opinion, introducing an additional figure or expanding the statistical analysis at this point would disrupt the narrative flow and risk confusing the reader.

(32) L141 "This is higher than the rate reported for a single young plant" Are you referring to the tomato plants used in the experiments? It might be helpful to include in the main text the natural click rate emitted by tomato plants, as this information is currently only mentioned in the Methods section.

See comment 4.

(33) L191 Is the main point here to convey that the plant playback effect remained significant even when the sound presentation frequency was reduced to 30 clicks per minute? The inclusion of the feeder element, however, seems to complicate the message. To simplify the results, moving the content from lines 185-202 to the supplementary materials might be a better approach. Additionally, what is the rationale for placing the sugar solution in the arena? Is it to maintain the moths' vitality during the experiment? Clarifying this in the methods section would help provide context for this experimental detail.

In this series of experiments, we manipulated four variables—single moths, ultrasonic click rate, arena configuration (from a two-choice design to an elongated enclosure), and the response metric (total egg counts rather than cluster counts)—to evaluate moth oviposition under more ecologically realistic conditions. We demonstrate the system’s robustness and validity in a more realistic setting (by tracking individual moths, counting single eggs, etc.).

As noted in the text, feeders were included to preserve the moths’ natural behavior and vitality. We have further clarified this in the revised manuscript.

(34) L215 Is the click presentation frequency 30 or 60 per minute? Since Figure 3 illustrates examples of moth movement from the experiment described in Figure 1, it might be more effective to present Figure 3 when discussing the results of Figure 1 or to include it in the supplementary materials for better clarity and organization.

See comments 1 and 4. As mentioned in the above

(35) L291 Please provide a detailed explanation of the experiments and measurements for the results shown in Figure S3 (and Figure S2). If the multi-modal hypothesis discussed in the study is a key focus, it might be better to include these results in the main results section rather than in the supplementary materials.

Thank you for this suggestion. Figure S2 was removed, see comments above. We’ve added now the context to figure S3.

(36) L303 It might be helpful to include information about the relationship between the moth species used in this study and tomato plants somewhere in the text. This would provide an important context for understanding the ecological relevance of the experiments.

Thank you for the suggestion; we have clarified the text accordingly.

(37) Table 1 The significant figures in the numbers presented in the tables should be consistent.

Thank you for the suggestion; we have clarified the text accordingly.

(38) L341 The text mentions that experiments were conducted in a greenhouse, but does this mean the arena was placed inside the greenhouse? Also, the term "arena" is used - does this refer to a sealed rectangular case or something similar? For the sound presentation experiments, it seems that the arena cage was placed inside a soundproof room. If the arena is indeed a case-like structure, were there any specific measures taken to prevent sound scattering within the case, such as the choice of materials or structural modifications?

Here, “arena” refers to the plastic boxes used throughout this study. In this particular experiment, we presented plants alone—reflecting ongoing debate in the literature—and used these trials as a baseline for our subsequent sound-presentation experiments, during which we measured sound intensity as described in the Methods section. All sound-playback experiments were conducted in sound-proof rooms, and acoustic levels were measured beforehand—sound on the control side fell below our system’s detection threshold.

(39) L373 "resister similar to the speaker" Could you explain it in more detail? I think this would depend on the type of speaker used-particularly whether it includes magnets. From an experimental perspective, presenting different sounds such as white noise from the speaker might have been a better control. Was there a specific reason for not doing so? Additionally, the study does not clearly demonstrate whether the electric and magnetic field environments on both sides of the arena were appropriately controlled. Without this information, it is difficult to evaluate whether using a resistor as a substitute was adequate.

Thank you for this comment. We have now addressed this point in the Discussion. We acknowledge that we did not account for the magnetic field, which might have differed between the speaker and the resistor. We agree that using an alternative control, such as white noise, could have been informative, and we now mention this as a limitation in the revised Methods.

(40) L435 60Hz? The representation of frequencies in the text is inconsistent, with some values expressed in Hz and others as "clicks per second." It would be better to standardize these units for clarity, such as using Hz throughout the manuscript.

We agree that this is confusing. We reviewed the text and made sure that when we addressed click per second, we meant how many clicks were produced and when we addressed Hz units it was in the context of sound frequencies.

(41) L484 "we quantified how many times each individual crossed the center of the arena" Is this data being used in the results?

Yes. Mentioned in the text just before Figure 3. L220